# Long-term responses of Icelandic Arctic foxes to changes in marine and terrestrial ecosystems

Fanny Berthelot[1]*, Ester R. Unnsteinsdóttir[2], Jennifer A. Carbonell Ellgutter[3], Dorothee Ehrich[1]

**1** Department of Arctic and Marine Biology, UiT -The Arctic University of Norway, Tromsø, Norway, **2** Department of Zoology, Icelandic Institute of Natural History, Reykjavik, Iceland, **3** Department of Natural Sciences and Environmental Health, USN-University of South-Eastern Norway, Bø i Telemark, Norway

* fanny.berthelot44350@gmail.com

**Data Availability Statement:** All relevant data are within the paper and its Supporting information files.

## Abstract

The long-term dynamics of predator populations may be driven by fluctuations in resource availability and reflect ecosystem changes such as those induced by climate change. The Icelandic Arctic fox (*Vulpes lagopus*) population has known major fluctuations in size since the 1950s. Using stable isotopes analysis of bone collagen over a long-time series (1979–2018), we aimed at identifying the main resources used by Icelandic Arctic foxes during periods of growth and decline to assess if the variations in their population size are linked to fluctuations in the availability of resources. We hypothesized that (1) the decline in Seabird abundance was responsible for the decrease in the fox population; and (2) that the growth in the fox population combined to fluctuations in main resources would lead to an increase in intra-specific competition, ultimately leading to variations in their isotopic niches at the population scale. The isotopic composition of Arctic fox bones differed clearly between inland and coast. Stable isotopes mixing models suggested that marine resources and rock ptarmigans were the most important food source and highlighted a rather stable diet in coastal habitats compared to inland habitats where more fluctuations in dietary composition were observed. Coastal foxes had a broader niche than inland foxes, and there was more variation in niche size in the inland habitat. Our results tend to confirm that a general decline in seabird populations drove the decline in Arctic foxes, especially in coastal habitats. For the inland foxes, our results suggest that the lack of marine resources might have led to an increased use of ptarmigans especially during the most recent period.

## 1. Introduction

Many Arctic ecosystems are characterized by a close link to adjacent marine environments. Around 80% of the terrestrial Arctic lays within 100 km from the coast [1], and marine subsidies often play an important role in terrestrial food webs [2, 3]. At present, high latitude ecosystems are subject to rapid changes under the influence of global warming [4, 5]. These

**Funding:** The analyses and publication costs were covered by UiT – The Arctic University of Tromsø to Fanny Berthelot (master thesis costs and publication fund). The funders had no role in study design, data collection and analysis, decision to publish, or preparation of the manuscript.

**Competing interests:** The authors have declared that no competing interests exist.

changes may affect marine and terrestrial systems differently and thus modify the interplay between these neighboring ecosystems [6]. Warmer and longer summers may lead to increased primary production on land, and better conditions for some herbivores [7], whereas expanding boreal species can become serious competitors of arctic endemics [8]. Warming of the sea causes major changes in food webs and leads notably to declines of many seabird populations [9]. Several predator species, such as gulls (*Larus* spp.), skuas (*Stercorarius* spp.) or Arctic foxes (*Vulpes lagopus*) are able to exploit both marine and terrestrial resources, thereby constituting a link between both systems [3, 6]. As these opportunistic species are likely to reflect changes through complex ecological responses, their long-term monitoring is a challenging yet key process for a better understanding of the impacts of global warming on interconnected marine and terrestrial ecosystems [10].

The Arctic fox is a terrestrial mammalian predator endemic to the Arctic tundra. It has a circumpolar distribution and is abundant across most of its range, including most arctic islands. Different threats such as increasing competition with the red fox (*Vulpes vulpes*), habitat loss arising from climate change and declines in the abundance of key prey make Arctic foxes increasingly vulnerable in part of their range [11, 12]. Therefore, they have been chosen as a climate change flagship species by the International Union of Conservation of Nature [13]. They are also a target of international monitoring as their reliance on tundra ecosystems make them likely to highlight the impacts of climate change through species interactions [10, 14].

Arctic foxes have been attributed to two different resource use strategies that involve different reproductive patterns: lemming foxes and coastal foxes [15–17]. The first type behaves as an opportunistic lemming (*Lemmus and Dicrostonyx spp.*) specialist and adapts its breeding effort to the lemming cycle with large litters in peak years [3, 17, 18]. The second type is more generalist, and lives on Arctic islands deprived of lemmings such as Iceland, or Svalbard. These foxes feed on both marine and terrestrial resources [16, 19] and dispose of a more stable food supply, thus producing fewer cubs per year, but breeding more regularly [17]. While the population dynamics of lemming foxes primarily follow the cycles of their prey, coastal foxes are driven by trends in both terrestrial and marine resources [6, 17, 20].

Iceland makes up for a particularly interesting system when it comes to understanding Arctic fox population dynamics for several reasons. First, this specific coastal population is neither threatened by interspecific competition with the red fox, as Arctic foxes are the only canid species living in Iceland, nor by the collapse of rodent cycles because lemmings are absent from this island [19]. Second, the species being considered as a pest, hunting is known to be the main cause of mortality [21]. The hunting pressure is thought to be stable since 1950, and is regulated by Icelandic laws, which makes it an unlikely driver of the population dynamics, but the hunting statistics provide a long-term estimate of the population trends [20, 22]. The island is also free of infectious diseases that could potentially be fatal to foxes, such as rabies or distemper [23, 24].

Despite the stable hunting effort and the apparent absence of common ecological pressures, striking long-term fluctuations in Arctic fox numbers have been documented and attributed to variations in carrying capacity, likely driven by the distribution and fluctuations in abundance of prey [20, 22]. In addition, Hersteinsson & al. [25] found evidence for indirect climatic impacts through food availability. The hunting statistics indicated a decrease of the population from 1950 to 1970, which has been partly explained by a reduction in the rock ptarmigan (*Lagopus muta*) population [20, 26, 27]. This period of decline has been followed by a steady six-fold increase until 2008 which has been explained by a global rise in food abundance. Hersteinsson & al. [25] suggested that climatic variables such as the Sub-Polar Gyre, the North Atlantic Oscillation and summer temperature acted indirectly to increase the abundance of the

main preys. Based on prey remains at dens, Pálsson & al. [20] documented an increased predation on waders and geese during this growth period, as well as an important use of fulmars. Using stable isotopes analysis, Carbonell Ellgutter & al. [28] highlighted the importance of marine resources and suggested that they might have supported the increase in the fox population.

Recent population estimates have shown important fluctuations in foxes' numbers during the last decade, starting with a drastic drop reducing the population to half its size within 5 years (Fig 1). Since 2011, however, the population seems to have recovered. Stable isotopes of carbon and nitrogen reflect the main resources used by a consumer over a certain period and can thus provide a good insight into potential drivers of population changes [29]. Building on the study of Carbonell Ellgutter & al. [28], we used isotopic signatures of bone collagen over 40 years to identify the main resources used by Icelandic Arctic foxes during periods of population growth and decline. We investigate whether the recent fluctuations in the size of the population were associated with changes in main dietary components and assess whether they can be attributed to fluctuations in the availability of prey. As Carbonell Ellgutter & al. [28] highlighted that the steady growth in population was characterized by a rather constant diet and may thus have been driven by increasing seabird populations constituting a key resource, we hypothesized that the recent decline in several seabird species [9] might have negatively affected the foxes during the last decade, especially in coastal areas. In the inland, population fluctuations may be related to a switch in diet from rock ptarmigan to increased predation on geese [20]. The important growth of the fox population together with fluctuations in main resources may have also led to an increase in intra-specific competition. Unnsteinsdóttir & al.

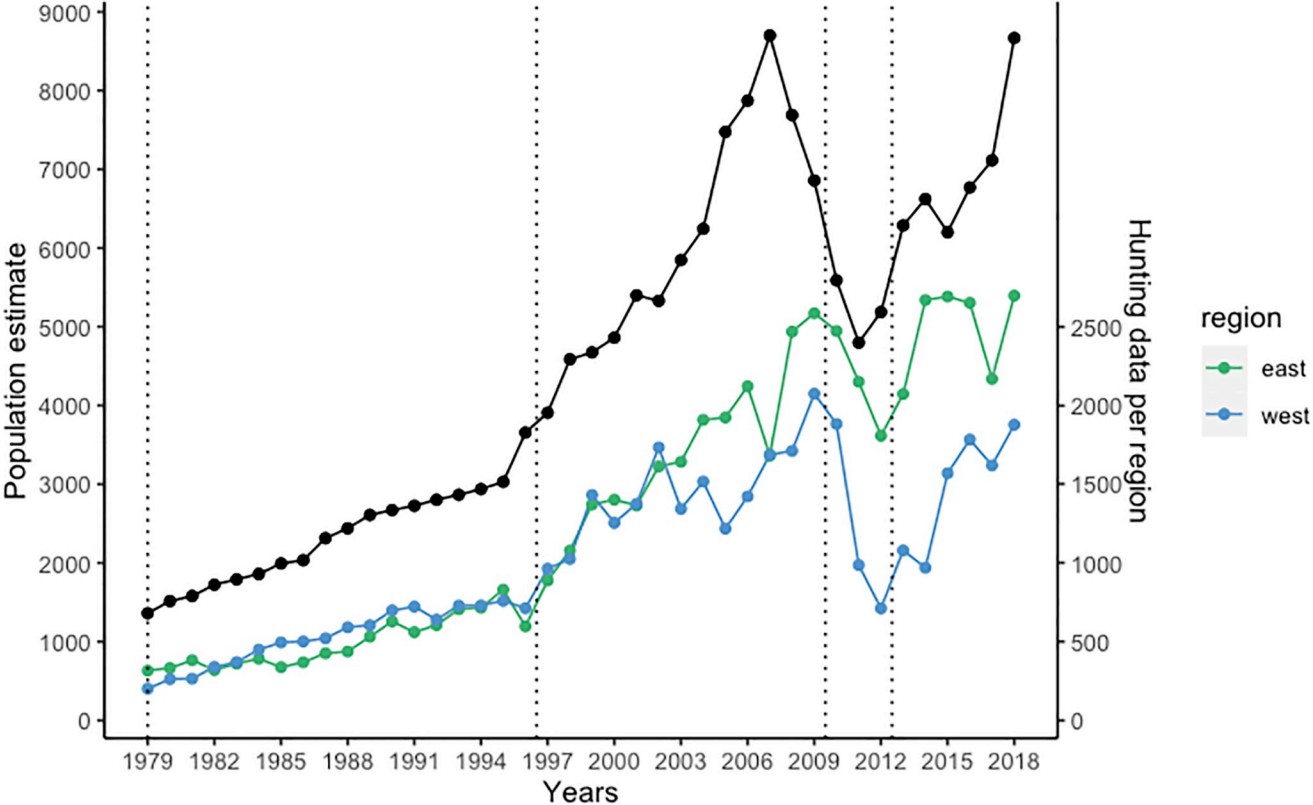

**Fig 1. Icelandic Arctic fox population estimates and hunting data per region from 1979 to 2018.**

[22] showed that density dependence is one of the main drivers of Icelandic Arctic fox population, pointing out that foxes adapt their territory sizes in response to variations in carrying capacity. Therefore, we hypothesized an increase in inter-individual variability in the diet with increasing population size, potentially leading to variation in the niche breadth at the population level. A better understanding of what caused the important variations this population experienced in recent years will be important for further management and hunting recommendations, and shed new light on how the interplay between changes in the marine and terrestrial ecosystems affect a flexible arctic predator.

The population estimates (in black) are based on age cohort analysis and hunting statistics [30] and are plotted along with the hunting data per region from 1979 to 2018 (note the different scales on the right and left axis). From a sample of aged foxes, life tables were created, and used to backtrack the age cohorts and create a model that predicts the proportion of each age group alive each year. The age cohorts were then accumulated to get the estimated number of foxes alive each year. Foxes culled further than 50km from the coast were referred as inland and were mostly coming from the East (in green), while foxes culled within 5km from the coast were considered coastal and were mostly culled in the West (in blue). The dotted lines delimitate the different period of growth and decline we are using in the study: 1979–1996; 1997–2009; 2010–2012; 2013–2018.

## 2. Material and methods

### 2.1 Study area and species

Iceland is located in the Atlantic Ocean, close to the Arctic Circle. Its climate is influenced by the Gulf Stream and the temperature is considerably higher than expected at this latitude. Monthly mean temperatures vary from -3 to 3˚C in January and reach 8 to 15˚C in July. Precipitation is high, ranging from 400 to 4000 mm annually (climateknowledgeportal.worldbank.org). The temperatures usually prevent the shorelines from freezing throughout the year. On average, the mean annual temperature has been above 2˚C during the last decades, whereas it was below 2˚C during the last century. Iceland is also free from pack-ice and thereby remains isolated [31].

Icelandic Arctic foxes can adopt two different resource use strategies, referred to as coastal and inland. Interior habitats are more subject to seasonal variations in temperature influencing the availability of resources. The resident rock ptarmigan (*Lagopus muta*) and migrating birds like waders, geese and passerines make up for most of the inland foxes' diet [16, 19, 25, 32]. The Western part of Iceland bears the most productive seashores, with a greater productivity than northern, southern and eastern Iceland combined [33]. It also supports most of the large seabird colonies that nest on the cliffs during summer [25]. Ice-free shores contribute to the stability of food supply [34], enabling coastal foxes to benefit from carrion from marine mammals, fish and marine invertebrates in addition to seabirds and some terrestrial preys [19]. All foxes can occasionally consume sheep (*Ovis aries*) and reindeer (*Rangifer tarandus*) carcasses, as well as cattle and horse carcasses that are used as baits by hunters [16]. Because of their more stable food resources, coastal foxes are thought to be more territorial whereas inland foxes are more mobile [19]. This subdivision is reflected in genetic differentiation between coastal foxes from the north-western part of Iceland and the foxes from the rest of the country [35]. Because of this clear distinction, we carried out all analyses addressing coastal and inland foxes separately.

### 2.2 Arctic fox samples

Fox mandibles were obtained from the collection of the Icelandic Institute of Natural History in Reykjavik, which consists of 12,200 mandibles from Arctic foxes culled from 1979 to

present. Legally killed fox carcasses are donated voluntarily by hunters from all over Iceland. All foxes have been aged by counting annual cementum lines of canine tooth roots [36] at Matsons laboratory (United States). To include both coastal and inland foxes and assure a sufficient sample size, we chose jaws of foxes culled in Nordur Isafjardarsysla (henceforth NIS) and Nordur Múlasysla (NMU), two counties respectively representing the Western productive seashores and the Eastern inland areas (Fig 2). Individuals from NIS were culled within 5km from the coast and can thereby be qualified as coastal whereas foxes culled in NMU were located further than 50km from the coast and are thus defined as inland. We included 106 samples from Carbonell Ellgutter & al. [28] culled in the same counties, along with samples coming from 5 other counties whose ecology was either similar to NIS or NMU (Ester Unnsteinsdóttir, personal communication), and culled within 5km from the coast or further than 50km from respectively. Altogether, this added up to a total of 256 samples, 127 of them being from inland areas and 129 being from coastal areas, covering a period of 40 years, from 1979 to 2018. The last 14 years, characterized by large fluctuations in abundance, were sampled more intensively. All the foxes analyzed were between one and two years old. As previous research showed that resource use as inferred from $\delta^{13}$C and $\delta^{15}$N does in general not differ between sexes [32, 37] male and female individuals have been chosen at random (S1 and S2 Tables).

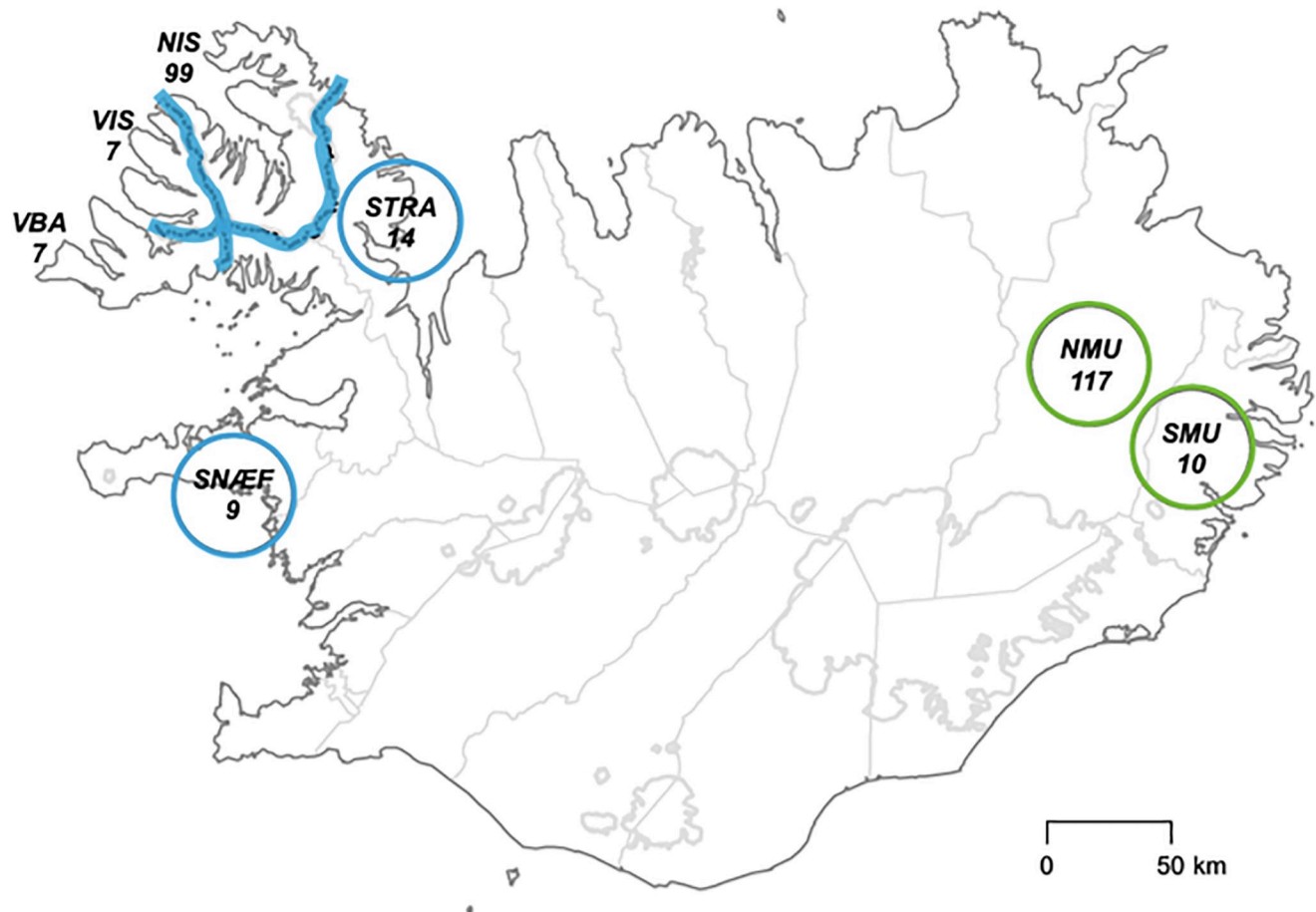

**Fig 2. Map of Iceland displaying the different culling locations.**

The isotopic composition of bone collagen has a slow turnover rate [38, 39]. It can reflect a lifetime average dietary intake although it is biased towards the period of greatest growth [3], which is until 8–9 months old for Arctic foxes [25]. Here, we will assume that the isotope signatures are representative of the diet of Arctic foxes during their first year of life, thereby reflecting their average resource use during this period. Collagen has been extracted from lower jaws following the same protocol as Carbonell Ellgutter & al. [28], based on a standardized method from Brown & al. [40] and modified according to Richard & Hedges [41]. Bone powder has been extracted with an electric drill and demineralized in a 0.25 M HCl solution for 6 days or until decalcification. After being filtered, rinsed and dried, the bone powder was grinded using a mixer mill (Tissuelyzer II, QIAGEN) before we proceeded to the lipid extraction. Lipid extraction was performed twice with 2:1 chloroform/methanol. The collagen samples were finally freeze dried before they were sent to analysis for stable isotopes of carbon and nitrogen at the Stable Isotopes in Nature Laboratory (SINLAB) at the Canadian Rivers Institute, University of New Brunswick. A Finnegan Delta V Plus isotope-ratio mass spectrometer was used coupled with a Finnegan Conflo IV and CE NC2500 Carlo Erba element analyzer. Stable isotope ratios were expressed using the standard $\delta$ notation in parts per thousand (‰) [42]. The international standards, i.e the Vienna PeeDee Belemnite for carbon and atmospheric air for nitrogen, were used as reference [42]. Data were calibrated using a three-point calibration curve and international reference material as standards (Nicotinamide, BLS, MLS, USGS61) were used for calibration while N2 and CH7 were used as check standards. Additional technical information can be found in S3 Table.

The quality of the inferred isotopic composition of the collagen samples was checked following Guiry and Szpak [43] using the atomic C:N ratio as quality criteria. Based on collagen composition reported by the authors and on relationships between $\delta^{13}C$ and C:N in our data we excluded data with an atomic C:N ratio exceeding 3.33, the highest ratio reported for mammalian collagen (see S4 Table for a justification of this threshold). We thus included 25 samples with an atomic C:N ratio between 3.28 and 3.33, which need to be considered with caution as they may be contaminated to a certain degree with non-collagen material. We excluded 23 samples with an atomic C:N ratios above 3.33.

Coastal areas are shown in blue, and inland areas in green. Culling locations are shown as a circle when hunting does not occur in the whole region. The number of foxes included in our dataset is specified under the abbreviation for each region. Reprinted from the Icelandic Institute for National History under a CC BY license, with permission from Hans H. Hansson, original copyright 2023.

## 2.3 Prey samples

Greylag goose (*Anser anser*) muscle and egg samples along with northern fulmar (*Fulmarus glacialis*) muscle samples have been provided by the Icelandic Institute of National History. Both muscle and egg samples have been prepared for stable isotope analysis following the method from Ehrich & al. [44]. Egg samples were dried and subdivided into a sample to analyze for $\delta^{15}N$, and a sample subjected to lipid extraction with 2:1 chloroform-methanol, before measuring $\delta^{13}C$. Muscle samples were cleaned with ethanol, dried, grinded in a mixer mill and subjected to chemical lipid extraction. Samples were analyzed for stable isotopes in carbon and nitrogen at SINLAB, along with the collagen samples. Additional prey signatures of ptarmigan, common eider (*Somateria mollissima*), wood mouse (*Apodemus sylvaticus*), golden plover (*Pluvialis apricaria*), whimbrel (*Numenius phaeopus*), sheep, horse, kittiwake (*Rissa tridactyla*), starfish (*Asteria rubens*), redshank (*Tringa totanus*), common snipe (*Gallinago gallinago*), and black guillemot (*Cepphus grille*) were obtained from Carbonell Ellgutter & al. [28].

## 2.4 Data analysis

The statistical analysis was performed using R version 4.0.4 [45]. We corrected both foxes and preys' raw $\delta^{13}$C values for the Suess effect, which consists in a depletion in $\delta^{13}$C in the biosphere driven by the input of $CO_2$ from fossil fuel since the Industrial Revolution [46]. To be consistent with Carbonell Ellgutter & al. [28], we followed the same method, i.e using a mean $\delta^{13}$C rate of change of -0.026% per year [47] and correcting all the $\delta^{13}$C isotopic ratios to levels, which correspond to the first year of the study (1979). The resulting correction was close to measurements obtained from terrestrial ecosystems [48].

The distribution of Arctic foxes' isotopic signatures with respect to their potential preys was assessed graphically by plotting individual values from both habitats. Prey values were corrected for isotopic discrimination, which corresponds to the amount of change in isotope ratios occurring as a resource is incorporated into the consumer's tissue [42, 49]. Since the discrimination factor for Arctic fox bone collagen has not been determined experimentally, we used a combination of the experimentally determined trophic discrimination values from Arctic fox blood cells (+0.49‰ and +2.56‰ for $\delta^{13}$C and $\delta^{15}$N respectively) [37], and a correction factor between red blood cells and bone collagen estimated for wolves by Adams & al. [50] (+2.6‰ and +0.8‰ for $\delta^{13}$C and $\delta^{15}$N respectively). The resulting estimate for the discrimination factor from the diet to bone collagen of Arctic foxes was used to adjust our prey signatures with: 3.09 ± 0.25‰ and 3.36 ± 0.37‰ for $\delta^{13}$C and $\delta^{15}$N respectively. This discrimination factor is an approximation and cannot fully account for the complexity of all the molecular processes involved in isotope discrimination during resource assimilation, thus some uncertainty about true discrimination remained.

Temporal changes in Arctic fox isotopic values were analyzed using generalized additive models (GAM) with the mgcv package to allow for non-linearity [51]. Changes in $\delta^{13}$C and $\delta^{15}$N have been modelled as a smooth function of the birth year of each individual. An interaction allowed to fit different changes over time for inland and coastal foxes, and their different means with respect to the two isotopes ratios were modeled as a fixed effect. We chose the restricted maximum likelihood method and used the default parameters of the package for both smoothing parameter and the number of basic functions. Both *gam.check* and *concurvity* functions were used to test for the fit of the model as recommended by Ross & al. [52] (S5 Table).

The proportions of different prey in Arctic foxes' diet over time were estimated using Bayesian stable isotope mixing models as implemented in the MixSIAR package [53] and were illustrated with boxplots. As mixing models perform best with few potential sources, the different prey items were grouped considering the similarity of their isotopic signatures and their ecological relevance. We created four distinct groups: cliff nesting seabirds (black guillemot, northern fulmar and kittiwake) called *seabirds*, eider ducks that had distinct isotopic signatures from the other seabirds were grouped with whimbrels to *marine resources*, the rock ptarmigan was kept as a focal species, and all other terrestrial prey (common snipe, greylag goose, goose egg, redshank, golden plover, wood mouse, horse, sheep) were grouped to *terrestrial resources*. Because of their more marine signature, whimbrels were placed with eider ducks, but the other wader species were placed in the *terrestrial resources*. To address potential changes in dietary composition, we defined four periods characterized either by growth or decline of the Arctic fox population (Fig 1). The mixing models were run separately for coastal and inland foxes. To account for the uncertainty considering the discrimination factor and because the results of mixing models can depend on how correct the discrimination value is [49], the analysis was repeated with the discrimination factor used by Carbonell Ellgutter & al. [28] (S6 Table, S1 Fig), in addition to the factor described above. We ran the analysis following

the MixSIAR manual recommendations, and did 1 000 000 or 300 000 000 MCMC replicates, (preceded by 500 000 or 1 500 000 burn-in), depending on when convergence was reached, and used a residual*process error [54]. The performance of mixing models being sensitive to the selection of priors [55], and as all priors are informative in a mixing context, we used priors based on the known dietary preferences of the foxes from both habitats, i.e. predominantly marine diet for coastal foxes and terrestrial diet for inland foxes [16, 32]. Thus, the prior attributed to coastal foxes was 2/3 of marine diet components (1/3 seabirds and 1/3 marine resources) and 1/3 terrestrial diet components shared between the two groups (1/6 rock ptarmigan and 1/6 terrestrial resources). For inland foxes the prior was opposite with 2/3 terrestrial diet components shared between the two groups and 1/3 marine diet components. The analyses were also run with uniform priors for comparison. The convergence of the MCMC estimations was assessed based on Gelman-Rubin and Geweke diagnostics, and we also inspected the correlation between different sources. In addition, we checked that the posterior distributions were unimodal. The isospace plots for the converging models are shown in S2 Fig.

The use of stable isotopes to infer a population's trophic niche width is increasing as isotopic niches are considered a good proxy for ecological niches [56–58]. For each period and habitat, we determined the isotopic niche breadth using standard ellipses containing 80% of the data as calculated by the SIBER package. The areas of those ellipses were computed using the Bayesian approach (SEAb, parametrized as detailed in Jackson et al. [59], S3 Fig). The ellipses being unbiased for sample size, it was possible to compare them even though the periods did not contain an equal number of individuals.

## 3. Results

The prey species showed the typical distinction between terrestrial species with lower $\delta^{15}$N and $\delta^{13}$C values and marine species with higher isotopic compositions for both carbon and nitrogen (Fig 3). The fox values were within the polygon delimited by the prey and covered the whole gradient from terrestrial to marine resources. Many coastal foxes had isotopic signatures that placed them close to the marine prey, whereas inland foxes were in general placed at the other end of the coast-inland gradient. Accordingly, both carbon and nitrogen isotopic compositions showed a significant difference between foxes of the two habitats, with an average difference of respectively 5‰ and 3‰ in $\delta^{13}$C and $\delta^{15}$N values between coastal and inland areas (Table 1).

Prey species were corrected with a discrimination factor resulting from the combination of the Arctic fox blood to diet discrimination from Lecomte & al. [37], and the blood to collagen variation of wolves from Adams & al. [50]. The prey isotopic composition corrected for trophic discrimination are plotted with their respective standard deviation.

For each model, the fixed effect of the difference between habitats is given (Habitat (inland)) as well as the estimates for non-linearity (edf—effective degree of freedom) for foxes from each habitat.

Significant effects are shown in bold

The GAM did not reveal significant trends in $\delta^{13}$C or $\delta^{15}$N over the study period (Fig 4). For coastal foxes, non-linear variations were detected (edf = 3.943; $p$ = 0.122) with fluctuations in the start of the study period and more stable and slightly higher values in recent years. For inland foxes, $\delta^{13}$C were stable over the study period. Regarding nitrogen, for coastal foxes, the values of $\delta^{15}$N were close to remain stable throughout the whole study period (edf = 1.644; $p$ = 0.598; Fig 4b) while more fluctuations were observed for inland foxes (edf = 4.537), with a noticeable increase at the end of the study period that was close to significant ($p$ = 0.063). The *gam.check* function showed full convergence for both models, as well as randomly distributed

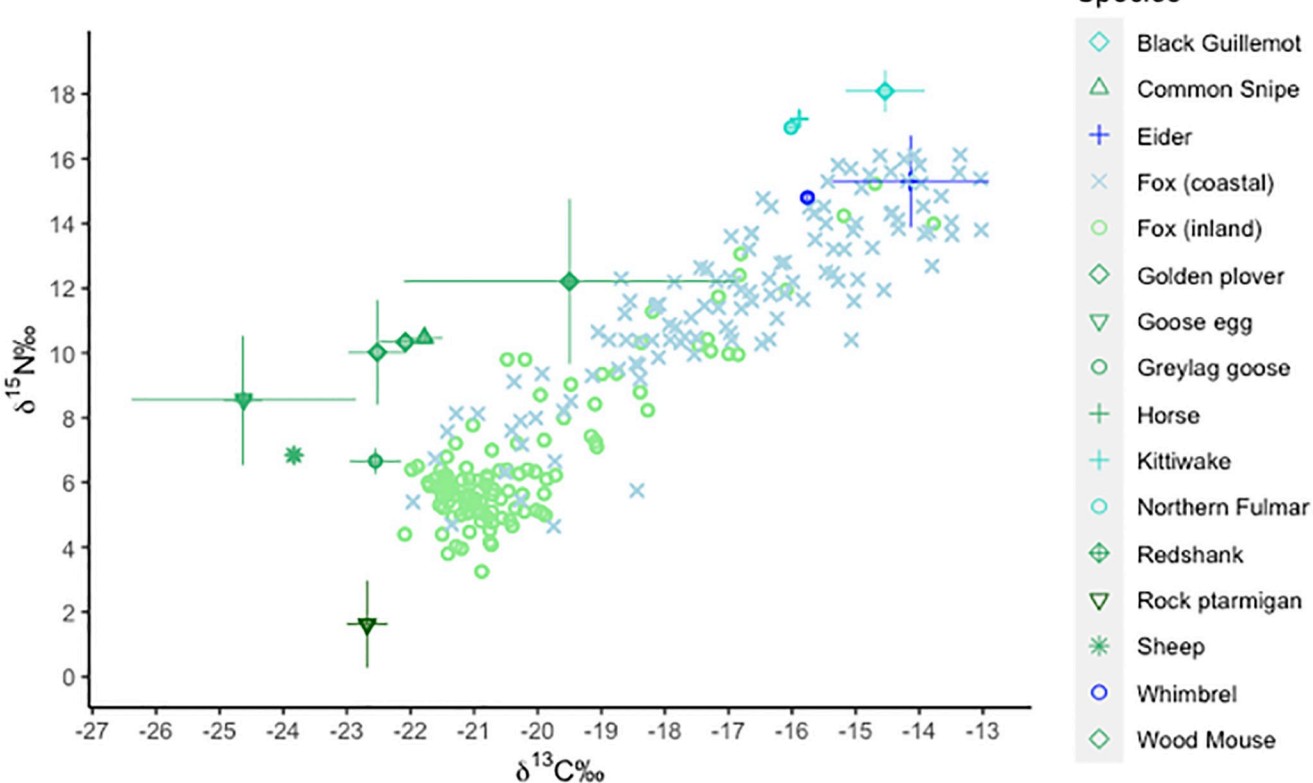

**Fig 3. Isotopic signatures (‰) of foxes from coastal and inland habitats plotted along with their potential prey species corrected for trophic discrimination.**

**Table 1. Parameter estimates from generalized additives models assessing the effect of birth year and habitat for (a) carbon isotopes ($\delta^{13}$C) and (b) nitrogen isotopes ($\delta^{15}$N) from bone collagen, along with their corresponding smooth terms values.**

| | Estimate | Std. Error | $t$ value | $P$ |
|---|---|---|---|---|
| (a) $\delta^{13}$C  Formula: $\delta^{13}$C ~ s(Year, by = Habitat) + Habitat | | | | |
| Intercept | -16.8597 | 0.1759 | -95.87 | **<2e-16** |
| Habitat (inland) | -3.3818 | 0.2498 | -13.54 | **<2e-16** |
| (b) $\delta^{15}$N  Formula: $\delta^{15}$N ~ s(Year, by = Habitat) + Habitat | | | | |
| Intercept | 11.7296 | 0.2297 | 51.07 | **<2e-16** |
| Habitat (inland) | -5.1021 | 0.3293 | -15.49 | **<2e-16** |
| | edf | Ref.df | F | $p$-value |
| (a) $\delta^{13}$C  Formula: $\delta^{13}$C ~ s(Year, by = Habitat) + Habitat | | | | |
| Coastal | 3.943 | 4.871 | 1.988 | 0.122 |
| Inland | 1 | 1.001 | 1.259 | 0.263 |
| (b) $\delta^{15}$N  Formula: $\delta^{15}$N ~ s(Year, by = Habitat) + Habitat | | | | |
| Coastal | 1.644 | 2.036 | 0.496 | 0.598 |
| Inland | 4.537 | 5.561 | 1.895 | 0.063 |

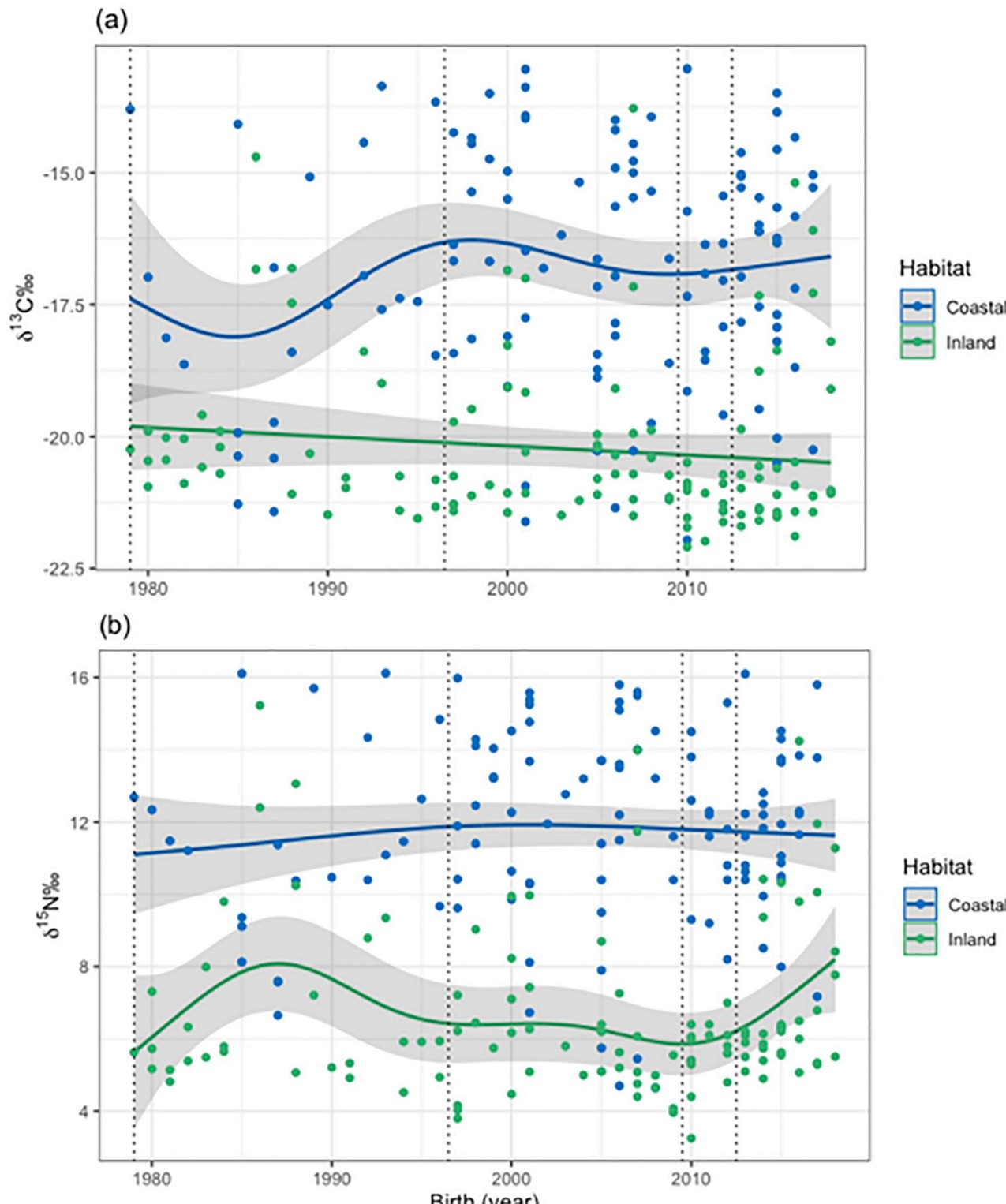

**Fig 4. Isotopic signatures in (a) $\delta^{13}$C and (b) $\delta^{15}$N of coastal and inland foxes plotted according to their year of birth.** Lines have been generated with generalized additive models, along with the 95% confidence interval.

residuals ($p > 0.05$ in both cases), thus confirming that the default parameters of the program were adequate. The *concurvity* function showed no evidence for concurvity between variables (S5 Table).

According to the results of the mixing models, the dietary composition of coastal foxes remained stable throughout the study period (Fig 5a) and the models was very close to converge with respect to the discrimination factor chosen (S6 Table, S1 Fig). As expected, marine diet components dominated for coastal foxes, but contrary to our expectations seabirds were less important that other marine resources. However, the posterior diet proportions of seabirds and alternative marine resources were highly correlated (-0.94), indicating that the distinction between these two prey groups was difficult to estimate reliably. This was reflected in considerable overlap in the credibility intervals of the dietary proportions estimated (Fig 5a). Among terrestrial preys, the rock ptarmigan was clearly the most important resource. Inland foxes had a somewhat more variable diet with respect to the different periods (Fig 5b) and the models also converged with both discrimination factors (S4 Table, S1 Fig). Ptarmigans were always the most used resource, but their proportion varied over time, representing nearly 70% of the diet during the decline phase. Marine preys were the second most important prey type, but their use decreased slightly during the decline phase. Other terrestrial resources were less important. The credibility intervals of seabirds did not exclude 0 in all periods, and this also applied to terrestrial resource during the two first periods. The results of models using a uniform prior were nearly identical. The MixSIAR analysis using the discrimination factor from Carbonell Ellgutter & al. [28] also provided similar results, with slightly clearer temporal fluctuations for inland foxes (S6 Table, S1 Fig).

As for the mixing models, the variations in the isotopic niche breadths estimated as standard ellipse areas were more important for inland foxes (Fig 6a). The niches kept a similar width until the decline phase when they shrank considerably. Coastal foxes' niches remained stable and overlapped during the different periods (Fig 6b). Moreover, the estimated niche areas of coastal foxes were overall greater than the ones of inland foxes (S3 Fig).

## 4. Discussion

### 4.1 Variations in resource use

Our results confirmed the previously described differences in diet between coastal and inland foxes, with coastal foxes having overall more marine isotopic signatures than inland foxes [28, 34]. The stable carbon isotope composition among inland foxes over the study period was combined with a slight increase in nitrogen istotopic composition in recent years. Coastal foxes, on the opposite, showed no statistical evidence for changes in $\delta^{13}$C and $\delta^{15}$N values, and the changes in $\delta^{13}$C they underwent during the beginning of the study had a low statistical support, and were likely due to the small number of individuals sampled during the first years of the study. These overall rather stable diets both at the coast and in inland habitats were in agreement with the estimations of dietary composition from isotopic mixing models. Both analyses also suggested some slight fluctuations, although not significant, in the inland. This was in agreement with the temporal changes in isotopic niche breadths, which showed important changes in the resource use of inland foxes while coastal foxes seemed to have a rather stable diet and constant niche width throughout the years.

In agreement with Carbonell Ellgutter & al. [28], who highlighted the importance of marine resources in the diet of Icelandic Arctic foxes, the mixing model analysis showed that marine preys were the main resource consumed by coastal foxes, and that they also were important to a certain degree for inland individuals. However, contrary to Carbonell Ellgutter & al. [28], we separated the marine resources in two groups and our results suggested that preys from lower

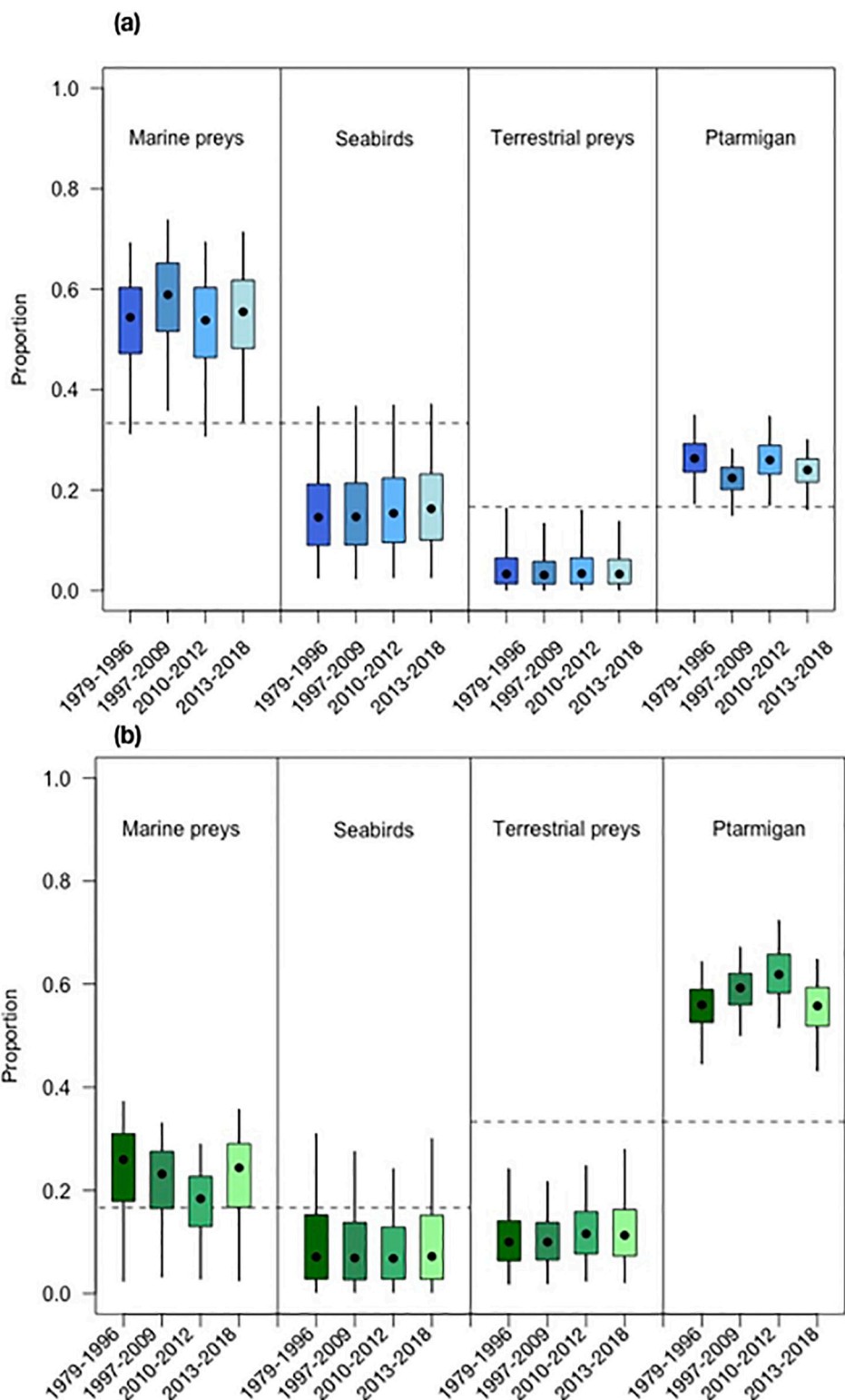

**Fig 5. Boxplots representing the proportion of different prey items in the diet of (a) inland and (b) coastal Arctic foxes during 4 different periods.** The box plots are based on the results of the MixSIAR analysis. The median and the confidence interval are also represented.

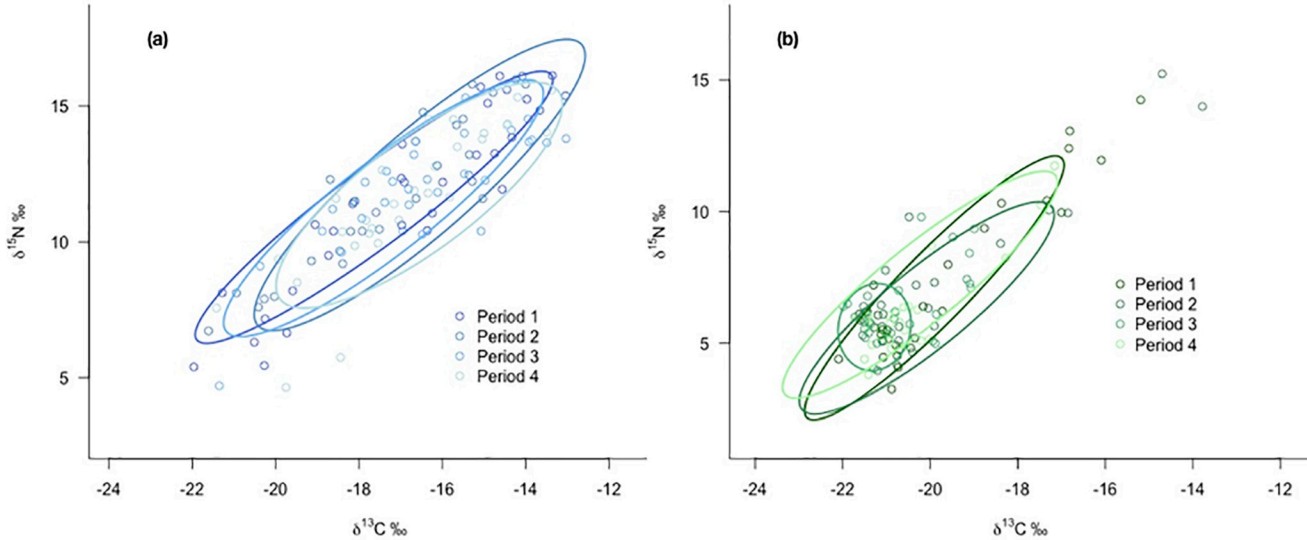

**Fig 6. Standard ellipses representing the isotopic niche breadths of (a) inland and (b) coastal foxes during four different periods.** Isotopic niches were calculated as standard ellipses areas containing 80% of the data with the "SIBER" R-package.

trophic levels such as common eiders and whimbrels could be more important than cliff-nesting seabirds. Although the separation between the two groups of marine resources is not strongly supported by the mixing model (high correlation between the posterior distributions and overlap of credibility intervals), it appears likely that marine resources like common eiders, which are the most common waterfowl available throughout the year and are widely distributed in coastal areas, could represent an important part of the diet of Icelandic Arctic foxes. However, predation on eiders in Iceland is thought to be lower than for other populations in other areas since their protection is one of the reasons for the Icelandic fox culling program [26, 60]. A high importance of common eider in Arctic fox diet is also contrary to findings from prey remains at dens [20], where fulmars were the most common species. This unexpected result could possibly be due to the fact that many of the coastal foxes were culled at eider colonies (Unnsteinsdóttir, personal communication).

Alternatively, whimbrels, which were included in our marine preys, could have been an important resource for foxes from both habitats as they are abundant and accessible during the breeding season [61]. Unfortunately, the grouping used in the mixing models analysis makes it hard to determine whether or not this particular wader species was more important than the others, and no information was found in literature to either support or contest this assumption.

The rock ptarmigan, also a year-round resident in Iceland, appeared to be the preferred terrestrial resource. They were especially important for inland foxes but were also consumed in significant amounts by coastal foxes (Fig 5b).

During the breeding season, waders and geese are increasingly available to foxes, and goose populations are especially increasing [20]. In contrast with Pálsson & al. [20] who suggested that these resources were the main preys available to inland foxes, our results suggested that these prey items were of minor importance. This confirmed the results from Carbonell Ellgutter & al. [28] who did not find support for an increased use of geese, despite isolating the greylag goose as a focal source in their mixing model analysis. The lack of species such as the pink-footed goose in our prey signatures could explain the apparent minor importance of this

group since this species has been shown to be especially important for Iceland Arctic foxes [16, 20]. The inclusion of terrestrial species like reindeer, which can occasionally be consumed by the foxes, would have given a more detailed picture of the diet of inland individuals, while the lack of marine species such as puffins (*Fratercula arctica*), crustaceans and other invertebrates might result in a underestimation of the proportion of marine resources, especially in the diet of coastal foxes. The use of freshwater prey by Arctic foxes in Iceland has not been reported in the literature [16, 19]. Although the isotopic composition of aquatic resources can be influenced by complex and diverse factors [39], the turnover rate of collagen reflects the lifetime dietary intake of the foxes. Thus, in the event of an occasional consumption of freshwater resources, it is unlikely that it would have affected our results. It is also important to note that the discrimination factor we used is an estimation as the fractionation has not been calculated for collagen in Arctic foxes yet. However, we assessed the robustness of our analysis using also the values from Carbonell Ellgutter & al. [28].

## 4.2 Driver of population change

As suggested by Unnsteinsdóttir & al. [22], the fluctuations in the population size of Arctic foxes likely result from changes in carrying capacity due to changes in abundance of main resources. Accordingly, the results of Carbonell Ellgutter & al. [28] showing a constant and important use of marine resources indicated that increasing populations of seabirds could have been a major driver of the long population increase of the foxes from the 1980's. For coastal foxes, our results showed a stable diet composition over the whole study period, despite considerable fluctuations in population size (Fig 1). This is also consistent with a bottom-up regulation of the fox population. Indeed, after a period of increase for many seabirds in Iceland up to the turn of the millennium, populations stabilized and some dramatically declined during the last decades [62, 63] when the numbers of foxes hunted in coastal areas started to fluctuate considerably. Breeding failure of several species was observed in 2005, while in 2010–2011 Puffin reproduction failed totally [64]. This was attributed to a lack of sandeel (*Ammodytes marinus*), a major resource for many seabird species [64, 65] notably for Fulmars [66], a key prey of Arctic foxes [20].

The strong decrease in fox numbers in 2011–2012 that was more pronounced for the coastal population might be a direct consequence of this event. Moreover, our results suggested that Common Eider is likely to be an important prey for Arctic foxes. Their populations stabilized and partly declined in Iceland in the 2000s after an overall increase in the end of the last century [67] and notably declined in Westfjords in western Iceland after 2000 [68]. This resource may thus also have contributed to flattening out and periodic decline of the coastal fox population.

For inland foxes, Ptarmigan were the main resources during the whole study period, but their population trends cannot really explain the increase in the Arctic fox population as many Icelandic populations showed declining trends in the last decades [69]. As suggested by Carbonell Ellgutter & al. [28] it is likely that the increase in marine resources also resulted to an improved situation for inland foxes during the growth period. This is supported by the fact that the crash of seabird populations together with declining Eider populations in 2010–2011 led to an abrupt decline in foxes hunted in the inland as well, although the decline was not as strong as for the coastal population. Notably fulmars were indeed breeding on cliffs far inland and preyed upon by Arctic foxes, but at present, these inland colonies are less active (ER Unnsteinsdóttir, personal observations). During these years, the marine input in the diet of the foxes declined and the proportion of ptarmigan increased. However, according to Fuglei & al. [69] ptarmigan populations were in the low phase in these years and could thus not

compensate the lack of marine resource for the foxes. After this decline phase, the diet of inland foxes returned to its previous state. The non-significant trend for an increase in nitrogen isotopic composition in the end of the study period might indicate increased use of geese and waders that experience positive population trends in inland areas [20], at least by some foxes–although such a diet shift was not detected by the MixSIAR analysis. In the future, inland foxes might adapt their diet to the changing resource situation shifting from declining Ptarmigan and seabirds to other terrestrial preys.

### 4.3 Population isotopic niche breadth

The high hunting pressure in Iceland leads to a high turnover in territorial foxes, and Unnsteinsdóttir & al. [22] suggested that Icelandic Arctic foxes engage in contest competition as they adapt their territory size in response to variations in carrying capacity.

Consequently, we predicted that Arctic foxes' niche breadth would vary over time, but found no support for this hypothesis in the coastal habitat. Although the diet of coastal foxes seemed not to vary over the study period, one could have expected that the decline in the availability of seabirds would have led to a narrower niche breadth or to a shift in the diet. The apparent consistency in their isotopic niche at the population level could hide some variations at a finer scale—the individual scale.

The results from isotopic niche analysis suggested that coastal foxes globally have broader niches than inland foxes. In previous research, Dalerum & al. [34] pointed out the same phenomenon and suggested that these wider niches were due to a diversification of individual strategies likely dictated by the local abundance of resources. The heterogeneity of coastal areas could lead to increased individual specialization of some foxes, especially since coastal foxes are more territorial [19]. This assumption would support that the fluctuations in carbon isotope composition observed among coastal foxes are likely to be influenced by individual variations in the diet rather than a global shift in the resource used at the population scale. This specialization would be a way of reducing the potential dietary overlap among foxes, in response to an increasing intra-specific competition pressure [70].

Inland foxes on the contrary showed more variations in isotopic niche space over time, as well as a marked reduction in their niche breadth during the period of decline. This fits with the increased use of rock ptarmigan suggested by the mixing model results during the decline period. In previous research, Hersteinsson [19] showed that the number of occupied fox dens was positively correlated to ptarmigan abundance, highlighting the importance of this prey item.

The reduction in the niche space matches the years of low productivity of seabirds, and illustrates the low availability of these resources, thus narrowing the niche. As the fox population recovered, the niche size widened, which could indicate that some marine species were available again.

## 5. Conclusion

Both marine and terrestrial ecosystems in Iceland are at present changing under the direct and indirect effects of climate change. Our results showed how the Arctic fox, a generalist top predator that uses different resources in western (coastal) and eastern (inland) Iceland, reacted to changes in resources availability. Coastal foxes that benefit from the productive seashores of western Iceland exhibited a constant marine dominated diet over the 40 years of our study. When seabird populations experienced reproductive failure, the fox population declined, probably because there were no alternative resources accessible. Arctic foxes are indeed genuine generalists able to exploit a wide variety of resources. As the seabird species, the foxes are

experiencing the profound changes in the marine food web related to lower reproduction of the sandeels that is related both to ocean warming and to structural changes of the marine food web [62, 64]. Inland foxes may have changed their diet more and therefore possibly experienced a somewhat less dramatic population decline. From being a stronghold of the Icelandic Arctic fox, with climate change the coastal areas may become a habitat with less reliable resources, whereas the prey basis in the inland areas may become more productive.

## Supporting information

**S1 Fig. Boxplot representing the proportion of different prey items in the diet of (a) inland and (b) coastal Arctic foxes using the discrimination factor from Carbonell Ellgutter & al. [28].**
(JPEG)

**S2 Fig. Isospace plots generated by MixSIAR on the convergent runs for (a) coastal and (b) inland habitats.** The discrimination factor used for this model was based on a combination of the fractionation values of Arctic fox blood from Lecomte & al. [37], and the blood to collagen variation of wolves from Adams & al. [50].
(JPEG)

**S3 Fig. Bayesian standard ellipse area determined with the SIBER package for (a) coastal and (b) inland foxes over the four periods. Parametrized as detailed in Jackson & al. [59].**
(PNG)

**S1 Table. Year of birth, location and sex of the foxes used in this study.**
(TIF)

**S2 Table. Extractions.** Extraction of Carbonell Ellgutter & al. [28] along with the one carried out in the present study.
(TIF)

**S3 Table. Technical details: Stable isotopes standards values.**
(XLSX)

**S4 Table. Raw isotopic data and collagen quality criteria.** Guiry and Szpak [43] recommend using a threshold of 3.28 in atomic C:N ratio to assess the quality of modern collagen samples of mammals. According to their study, samples with higher values result in a significant correlation between atomic C:N and $\delta^{13}$C, and $\delta^{13}$C values for these samples may be contaminated by non-collagen material. Among our samples, 48 had C:N values > 3.28. Not to exclude excessively many samples we decided to use a threshold of 3.33, that corresponds to the maximum ratio determined based on amino-acid composition by Guiry and Szpack [43]. Including samples up to this threshold resulted in a significant correlation between C:N and $\delta^{13}$C and we may thus have included samples with slight contamination. However, given the coefficient of the regression of $\delta^{13}$C against C:N was -8.13, a difference in C:N of 0.05 (3.28 to 3.33) would only lead to a change in $\delta^{13}$C of 0.4, corresponding to less than 0.2 standard deviations of our $\delta^{13}$C values. The bias this might have introduced would thus be negligible.
(XLSX)

**S5 Table. Outputs from (a) *gam.check* and (b) *concurvity* functions to test for the fit of the generalized additive models.** Estimated with the default parameters of the *gam* function in the mgcv package.
(XLSX)

**S6 Table. Overview of the runs performed in MixSIAR.** Foxes were analyzed separately depending on their habitat and time was included as a categorical (four periods) covariate. Coastal and inland foxes were analyzed separately. We used two different discrimination factors. The first was the one used in Carbonell Ellgutter & al. [28], and the second one was the addition of the values from Arctic fox blood from Lecomte & al. [37] and the variation from blood to collagen of wolves from Adams & al. [50]. We also ran the models with uniform priors (P1) and with informative priors (Pinf). Convergence was assessed based on the Gelman-Rubin and Geweke diagnostics. The detailed results of the Geweke diagnostics are shown in percentage of number of variables outside +/- 1.96 per chain. The maximal correlation between two sources is given, and sources are abbreviated as Ptarmigan–P, Alt. terrestrial–T, Alt. Marine–M, Seabirds–S. Models which had satisfactory convergence diagnostics are shown in bold.
(XLSX)

## Acknowledgments

The authors would like to thank Sissel Kaino for her help in the lab.

## Author Contributions

**Conceptualization:** Fanny Berthelot, Dorothee Ehrich.

**Data curation:** Fanny Berthelot, Jennifer A. Carbonell Ellgutter.

**Formal analysis:** Fanny Berthelot.

**Funding acquisition:** Dorothee Ehrich.

**Investigation:** Fanny Berthelot.

**Methodology:** Fanny Berthelot, Jennifer A. Carbonell Ellgutter.

**Resources:** Ester R. Unnsteinsdóttir.

**Supervision:** Ester R. Unnsteinsdóttir, Dorothee Ehrich.

**Writing – original draft:** Fanny Berthelot.

**Writing – review & editing:** Fanny Berthelot, Ester R. Unnsteinsdóttir, Dorothee Ehrich.

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
