## [Decision Letter · Decision Letter 0]

6 Mar 2023

PONE-D-23-03420Long-term responses of Icelandic Arctic foxes to changes in marine and terrestrial ecosystemsPLOS ONE

Dear Dr. Berthelot,

Thank you for submitting your manuscript to PLOS ONE. After careful consideration, we feel that it has merit but does not fully meet PLOS ONE’s publication criteria as it currently stands. Therefore, we invite you to submit a revised version of the manuscript that addresses the points raised during the review process.

Both reviewers highlighted the interest of this work. They present a set of recommendations, including critical points that need to be revised to allow the publication of the manuscript. Reviewer 2 insists on listing all raw data and applying quality criteria. I concur with reviewer 2 that not only modern or sub-modern samples may be contaminated, but we also need to push up our isotopic studies on modern and past ecosystem to the same standards. This includes giving full access to the raw data, as well as complete analytical procedures and full description of calibration and standards. It is also mandatory to provide such information to comply with the FAIR principles for scientific data management included in the PLOS ONE policies. Reviewer 2 indicates several points and literature sources that will be helpful to the authors. There is a need to specify the TEF values and the collagen-to-diet isotopic offset. It is not unusual to see some confusion between collagen-to-collagen and collagen/other tissue-to-diet isotopic offset in modern and past ecosystem trophic studies. As mentioned by reviewer 2, the Suess effect is not linear, and several papers provide detailed data or formula to take into account the exponential tendency of the change in atmospheric ^13^C amounts (e.g. Dombrosky 2020, The Holocene). Reviewer 1 states relevant questions and comments about the application of the mixing model. Altogether, addressing the comments of both reviewers through a major revision process will help the authors to enhance the impact of their study. 

We look forward to receiving your revised manuscript.

Kind regards,

Dorothée Drucker

Academic Editor

PLOS ONE

2. Please include a complete copy of PLOS’ questionnaire on inclusivity in global research in your revised manuscript. Our policy for research in this area aims to improve transparency in the reporting of research performed outside of researchers’ own country or community. The policy applies to researchers who have travelled to a different country to conduct research, research with Indigenous populations or their lands, and research on cultural artefacts. The questionnaire can also be requested at the journal’s discretion for any other submissions, even if these conditions are not met.  Please find more information on the policy and a link to download a blank copy of the questionnaire here: https://journals.plos.org/plosone/s/best-practices-in-research-reporting. Please upload a completed version of your questionnaire as Supporting Information when you resubmit your manuscript.”

5. We note that Figure 2 in your submission contain [map/satellite] images which may be copyrighted. All PLOS content is published under the Creative Commons Attribution License (CC BY 4.0), which means that the manuscript, images, and Supporting Information files will be freely available online, and any third party is permitted to access, download, copy, distribute, and use these materials in any way, even commercially, with proper attribution. For these reasons, we cannot publish previously copyrighted maps or satellite images created using proprietary data, such as Google software (Google Maps, Street View, and Earth). For more information, see our copyright guidelines: http://journals.plos.org/plosone/s/licenses-and-copyright.

Reviewers' comments:

Reviewer's Responses to Questions

**Comments to the Author**

1. Is the manuscript technically sound, and do the data support the conclusions?

Reviewer #1: Yes

Reviewer #2: Yes

2. Has the statistical analysis been performed appropriately and rigorously? 

Reviewer #1: Yes

Reviewer #2: Yes

3. Have the authors made all data underlying the findings in their manuscript fully available?

Reviewer #1: Yes

Reviewer #2: No

4. Is the manuscript presented in an intelligible fashion and written in standard English?

Reviewer #1: Yes

Reviewer #2: Yes

5. Review Comments to the Author

Reviewer #1: The manuscript discusses changes in the diet and trophic niches of Arctic foxes over a period of nearly 40 years. The authors examined the stable δ13C and δ15N isotope values from the bone collagen of the foxes. The study is very informative and shows very well the changes in diet over the last 40 years and the relationship between foxes and prey availability. However, I have some major and minor comments which should still be considered:

Major comments:

1. Line 127 f: The authors write that the populations were determined using "age cohort analysis and hunting statistics". Could this be a bit more in-depth? What is the relationship between population size and number of kills? Figure 1 gives the impression that the population is always twice the number of kills. I think this is too simplistic. I would ask the authors to elaborate on this.

2. Line 183 ff: The authors write that they extracted collagen according to a specific protocol. It would be desirable to say more about this. Please add a sentence or two describing the steps involved. This will help to classify the values obtained.

3. Line 185: Please enter the laboratory specifications here. This helps to classify how accurate the measurements are. Information about the standards used is also missing.

4. Line 193 ff: This section describes the prey samples. As far as I know, only muscle and egg samples are taken. It would also be important to briefly describe the method of preparation. Can muscle or egg isotopes be compared to collagen isotopes without calibration? If not, the formula for calibration must be mentioned here.

5. Line 214 f: I am not sure with these numbers if this is the correlation of collagen to muscle, or the TEF (Trophic Enrichment Factor) value, or the combination. Please make this clearer. The reference to Fig. 3 is even more confusing as it seems to include TEF values as well. Please clarify what your TEF values are and what the correlation values are.

6. Line 256 ff: Why did the authors use only 300,000 MCMC replicates when the cited manual recommends 1,000,000 MCMC for publications? I also know from many other publications using MixSIAR that 1,000,000 MCMC replicates is the standard size to get a clean running model. In addition, the results of the error diagnostics (Geweke & Gelman-Rubin) to evaluate the usability of the model are missing. These need to be included.

7. Line 269 ff: This section describes how SIBER was used. However, what is missing is the information on how the SEA was calculated. Did the authors use the 40% SEA, which is the core niche, or a 95% SEA, or some other SEA? This information needs to be added. The other niche metrics (e.g., TA and other layman metrics) that can be calculated with SIBER would also be desirable. SIBER also calculates SEAb (Bayesian SEA), which can be used to compare populations and time periods. The different sample sizes do not play a decisive role for this analysis. I strongly recommend including these calculations.

Minor comments:

- Citation in the text: When referring directly to a study, do not say “As [28] showed …” but the name followed by the number. If the authors use Citavi or Endnote, this can be fixed very quickly: Highlight the citation, click Edit, and set it to “display as author(year).

- Line 178: What did the authors mean with “one collagen”?

- Notation of δ13C and δ15N: Please use the international notation for δ13C and δ15N values. The number is superscripted, and the delta is italicized.

- Line 260: Did the authors run the model without priors and is it very different from the model with priors? Maybe the authors can put something like that in the SI.^

- Line 354: The authors should add a small paragraph here about how to deal with missing resources. The selected prey resources cannot reflect 100% of the Arctic fox diet. There may be a resource that was not considered that completely changes the picture. This should be briefly discussed.

- Line 444: This is where the SEAb (or SEAc) comparison mentioned in point 7 would fit in very well to support the statement about the size of the niche.

Reviewer #2: This manuscript was a pleasure to read and I would like to commend the authors on what appears to be an exciting study. That said, I recommend the following changes, which are aimed at enhancing the paper’s ecological and isotopic background and clarifying language around some interpretations. Note that I consider compliance with Comment's 1 and 2 to be critical to the paper's publishability.

General Comments

1) Methods Section. There is an essential, non-negotiable need to include reference to established collagen quality criteria (QC) for bone collagen isotopic compositions. This means reporting and referencing atomic C-to-N ratios (Atomic C:N, hereafter just “C:N”) and the carbon and nitrogen concertation data (%C and %N) that are used calculate them. I cannot stress the importance of ensuring that these data are included for all samples enough. This is the primary means by which the data quality all bone collagen samples (modern or archaeological) is evaluated, a convention that has been widely accepted by the those working with bone collagen stable isotope research for decades. In that context, not only is it key that you include these QC data to follow rudimentary standards of data hygiene practices, but including these QC data will also likely be key for ensuring more interdisciplinary engagement with your study. For instance, many broader meta-analyses (of the kind that are increasingly making big discoveries in retrospective animal isotopic compositions studies today) would exclude data that has no associated QC information. Below, I have offered a brief overview of C:N ratios with reference to the some of the nuances of which approach is best suited for work with modern samples.

1a) C:N ratios have long been a widely accepted standard across disciplines (as per the 3500+ citations of these key QC criteria papers: DeNiro, 1985, Ambrose, 1990) for ensuring that the isotopic compositions (especially for δ13C) of collagen are not altered by contamination. While these QC criteria were developed initially for ancient samples, they have long been used in contemporary ecology and have recently been revised (Guiry and Szpak, 2020) to further improve their ability to ensure data from modern samples are more accurate. In a nutshell, unlike, more commonly analyzed materials from modern specimens, such as muscle, collagen has a highly conserved, and specific Atomic C:N across taxa, which can provide a robust marker for contamination detection. For collagen from mammalian tissues, Atomic C:N that falls outside the 3.00-3.28 range (see Guiry and Szpak 2020) are contaminated with non-collagenous materials (typically lipids or other proteins) that have been incompletely removed from the bone sample during cleaning and collagen extraction. As these materials have isotopic compositions that can differ drastically (and unpredictably) from that of collagen, samples that fall outside of the acceptable range of Atomic C:N should be discarded, or used with caution and a clear caveat. I should note that, even within the acceptable Atomic C:N range, there can be meaningful impacts of on δ13C. If you’re worried about this, you can test for it by plotting your δ13C against Atomic C:N and looking for relationships. Important notes:

i) Atomic C:N (as opposed to molecular C:N) is used for collagen C:N work by widely accepted convention and is calculated as follows: (%C/%N)x(14.007/12.011).

ii) For dataset of the size you are presenting here, these QC data (i.e., C:N, %C, and %N) are typically provided in an ESM in the same table as your isotopic compositions for individual samples (see Comment 2 below).

2) Results Section. I do not see your raw data included in the manuscript or its supplements. The summary averages presented in your tables cannot stand in place of providing the raw data. Unlike ecology, where in the past original data have sometimes not published, it has become a fundamental requirement for studies using bone collagen data that all isotopic compositions and quality control metrics data are provided so that readers can assess the data for themselves and conduct inter-study comparisons. Globally, in the past 35 years, hundreds of thousands of isotopic compositions from extracted, purified bone collagen have been published in full in the archaeological, historical ecological, and paleontological literatures and data from previous studies are routinely used for comparative purposes to support interpretations in new studies. It is critical that we make our data as freely available and useful to future researchers as possible. A key reason for this is that isotopic analyses are a destructive process – the samples we cut from bones and then analyse can never be analyzed again. It therefore behoves us all to ensure that our data can contribute to wider discussions in the future. If you do not provide δ13C and δ15N values from extracted bone collagen, future studies will not be able to link datasets with yours – this may serve to discourage inclusion of your work in future debates. In other words, because isotopic analyses destroy unique, un-replaceable samples we have an ethical responsibility to make all raw data freely available. It is there for an essential requirement that all isotopic data used in this paper is made available to your readership in a supplement.

Specific Comments

3) Fig 1 caption. Please outline what the blue and green dots show in your caption.

4) Lines 184-185. More information is need here, particularly with respect to analytical reporting. Your collagen extraction methods derive from the archaeological and paleontological literatures, and this make sense as these are discipline areas with the most experience working out the nuances of preparing and analysing bone collagen isotopic compositions. However, you have stopped short of what current best practices in this (and other isotope- focused) disciplines require. In that context, I encourage you to consider following current best practices for reporting on bone collagen work as outlined by Szpak et al. (2017). Note that most of this can be done in an electronic supplementary materials section.

a) Explicitly note how data were calibrated (e.g., a one-, two-, or more-point calibration curve).

b) With respect to the standards you have used you should note:

i) what these are,

ii) which are check standards and which are calibration standards,

iii) the accepted or long-term average observed isotopic composition for each,

c) Report tables with the “n=”, means (for check standards), and standard deviations (for calibration standards and check standards) for analyses run in each analytical session.

d) Report tables with the “n=”, individual values, means, and standard deviations for all sample replicates.

e) Consider calculating uncertainty calculation following recommendations from Szpak et al. (2017).

5) Lines 204 to 209. You have glossed over a great deal of complexity here and it would be useful for readers, especially those who may not have an isotope background, to see explicit references to some of the sources of uncertainty built into all these data treatment variables as follows:

a) How have extraction methods for non-bone collagen tissues been dealt with in terms of potential for lipid contamination? What steps have you taken to ensure the best possible comparability between these data types?

b) As you mention, you don’t have a direct isotopic offset for fox bone collagen-to-diet and you have come up with a best guess based on stacked inter-tissue offsets. I think perhaps even adding an explicit, brief note that there are uncertainties that can’t be accounted for here is worthwhile, particularly in light of the vary different processes that underly each of the offsets you have come up with. For instance, while there are large isotopic offsets between both the δ13C and δ15N of an animal’s collagen and its diet, the underlying processes behind these and the extent to which they have been studied differs in some big ways. Diet-consumer bone collagen shifts in δ15N result from trophic enrichment of 15N and this has undergone a lot of study. The offset value you have selected here is broadly in line with results from relevant meta-analyses. By contrast, diet-consumer bone collagen shifts in δ13C are not the result of large trophic enrichment shifts, but instead derive largely from intra-individual, inter-tissue differences that are inherently difficult to study (due to the slow turnover of bone collagen and length of time needed for controlled feeding studies). In that context, it is more challenging to see if the offset you have come up with for δ13C is in the right ballpark.

c) The Susse correction is not linear. Instead, the rate of change in our atmosphere’s δ13C is accelerating more and more with each passing year. Because this process continues to build momentum, constant-rate Suess corrections are not well suited for samples from recent decades. I encourage you to consider using a more nuanced approach (Clark et al., 2021) that considers rate of change in your Susse correction. Alternatively, if you can calculate that this rate of change is inconsequential in the context of your interpretations (which, given the wide spread of δ13C you have observed, is entirely plausible), you could simply add a caveat to that effect.

d) When and where are the samples used for baseline data from? How have you accounted for potential systematic changes (aside from the Susse effect) in baseline prey isotope compositions? Is it not possible that baseline prey isotopic compositions could vary through time due to changes in prey behavior or environmental conditions (climatic or anthropogenic) influencing relevant C and N cycles? Have you made an attempt to source prey baseline data over the same period of time for which you exampling foxes?

6) Line 221. For Figure 3, I suggest the following changes to improve legibility:

a) Reduce the integer interval for your vertical axis. An interval of, say, 2‰ might make this a bit less cluttered and easier to read.

b) Add the standard reference materials to relevant axis titles (i.e., AIR, VPDB).

c) Consider changing the color of one of the fox groups. For instance, if you were to make inland fox “x” symbols black, this would help them ‘pop’ a bit better. As it stands, in some areas of the plot, the data forms a cloud that is difficult to make sense of.

7) Lines 243-251. Somewhere (elsewhere would be fine if you prefer) you should deal with the potential for freshwater food use by: 1) acknowledging that freshwater foods represent a tremendous range of potential variation in source isotope values (for review see Guiry, 2019) and, 2) outline the extent to which such resources might have been available to foxes. I suspect, based on the fact that you haven’t touched on this as a possibility, that there is a low likelihood that foxes have had substantial access to freshwater resources. However, reference to the fact your aware of this as a potential interpretive issue would be useful for your readers who are not familiar with there are (and could therefore misunderstand this as an omission of potentially relevant baseline source data).

8) Line 310. “that” should be “than”. Note that, though I have seen several potential word choice issues, I have not been made a systematic attempt to comment on these. I suggest you go over the manuscript in detail prior to resubmission.

9) Line 346. Here and throughout the manuscript (e.g., line 379, 458) please be more specific about what kind of values you are referring to. It is not good practice to use vague terminologies like “nitrogen ratios”, “carbon ratios”, or “carbon signature”. In the case of “nitrogen ratios”, for instance, this is because this could technically refer to a ratio involving the concentration of nitrogen (i.e., %N, which should also be reported as a key QC metrice (see Comment 1 above) or nitrogen isotope compositions (i.e., δ15N). I understand what you mean here but it is important to use correct terminology to help avoid confusion among readers with less experience in this area (for a review on terminology see Coplen, 2011).

References

AMBROSE, S.H. 1990. Preparation and characterization of bone and tooth collagen for isotopic analysis. Journal of Archaeological Science 17: 431-451.

CLARK, C.T., CAPE, M.R., SHAPLEY, M.D., MUETER, F.J., FINNEY, B.P. & MISARTI, N. 2021. SuessR: Regional corrections for the effects of anthropogenic CO2 on δ13C data from marine organisms. Methods in Ecology and Evolution 12: 1508-1520.

COPLEN, T.B. 2011. Guidelines and recommended terms for expression of stable‐isotope‐ratio and gas‐ratio measurement results. Rapid communications in mass spectrometry 25: 2538-2560.

DENIRO, M.J. 1985. Postmortem preservation and alteration of in vivo bone collagen isotope ratios in relation to palaeodietary reconstruction. Nature 317: 806-809.

GUIRY, E. 2019. Complexities of stable carbon and nitrogen isotope biogeochemistry in ancient freshwater ecosystems: implications for the study of past subsistence and environmental change. Frontiers in Ecology and Evolution 7.

GUIRY, E. & SZPAK, P. 2020. Quality Control for Modern Bone Collagen Stable Carbon and Nitrogen Isotope Measurements. Methods in Ecology and Evolution.

SZPAK, P., METCALFE, J.Z. & MACDONALD, R.A. 2017. Best practices for calibrating and reporting stable isotope measurements in archaeology. Journal of Archaeological Science: Reports 13: 609-616.

6. PLOS authors have the option to publish the peer review history of their article (what does this mean?). If published, this will include your full peer review and any attached files.

Reviewer #1: No

Reviewer #2: No

---

## [Author Response · Author response to Decision Letter 0]

18 Jul 2023

Dear Editors, 

1. We did our best to follow PLOS one's requirement. 

2. We filled in the questionnaire on inclusivity with the information we had available and attached it in "other".

3. The funding information has been modified. 

4. This statement was a mistake and we would appreciate help to make our data available. 

5. We followed the guidelines for copyright issues and added the permission in the "other" attached files.

---

## [Decision Letter · Decision Letter 1]

10 Aug 2023

PONE-D-23-03420R1Long-term responses of Icelandic Arctic foxes to changes in marine and terrestrial ecosystemsPLOS ONE

Dear Dr. Berthelot,

Thank you for submitting your manuscript to PLOS ONE. After careful consideration, we feel that it has merit but does not fully meet PLOS ONE’s publication criteria as it currently stands. Therefore, we invite you to submit a revised version of the manuscript that addresses the points raised during the review process.

Both reviewers and I acknowledge the thorough revision of the manuscript by the authors. Some minor revisions should be done before complete acceptance of the paper.

I encourage the authors to consider the following points (line references according to the text without tracked changes):

please answer comment 2) of reviewer 1 and add the results of the statistical tests in table S6Line 201-202: Are all the standards quoted equivalent to international reference material such as USGS or is there any internal (=inhouse) reference material (which is totally fine)?Line 208-209 “*We thus included 25 samples with a C:N ratio between 2.28 *and 3.33” *while the authors convincingly argue about considering isotopic results on collagen with C:N atomic ratio up to 3.33, I do not see any argument to keep samples with C:N below 3.0 as recommended by Guiry &Szpak (not Szpack as in table S4) 2020 and could not spot any atomic C:N under 3.09  in your table S4 anyway. I guess you confused with the molecular C/N ratio of column J. Please specify that J column is a molecular C/N ratio and K column is an atomic C/N ratio (named C:N by convention), avoiding further confusion, and amend your main text accordingly. You may not necessarily refer to a specific literature source for the calculation of the atomic C/N ratio, since it corresponds to chemical convention (14.007 and 12.011 being the atomic weight of N and C; please note that initially C and N contents where measured by the Kjeldahl method giving automatically C/N atomic ratios that are still our landmarks today despite being provided with molecular ratio through the mass spectrometer facility).*Line 385-386: “*no statistical evidence for changes in nitrogen isotopic composition, and the changes in δ^13^C*” please replace by “ no statistical evidence for changes in *δ*^13^C  and  *δ*^15^N  values” for sake of consistencyLine 429-431: I guess it is where you consider missing resources as recommended by reviewer 1. Could you add a sentence about the consequences of missing terrestrial resources in your reconstruction, as you did for freshwater resources, to better address the point of the reviewer? In other words, beside t the pink-footed goose, which other terrestrial species contribution may have been over- or underestimated? 

After addressing these minor points, I will be glad to recommend the manuscript for publication and congratulate the authors for their work.

We look forward to receiving your revised manuscript.

Kind regards,

Dorothée Drucker

Academic Editor

PLOS ONE

Journal Requirements:

Reviewers' comments:

Reviewer's Responses to Questions

**Comments to the Author**

1. If the authors have adequately addressed your comments raised in a previous round of review and you feel that this manuscript is now acceptable for publication, you may indicate that here to bypass the “Comments to the Author” section, enter your conflict of interest statement in the “Confidential to Editor” section, and submit your "Accept" recommendation.

Reviewer #1: (No Response)

Reviewer #2: All comments have been addressed

2. Is the manuscript technically sound, and do the data support the conclusions?

Reviewer #1: Yes

Reviewer #2: Yes

3. Has the statistical analysis been performed appropriately and rigorously? 

Reviewer #1: No

Reviewer #2: Yes

4. Have the authors made all data underlying the findings in their manuscript fully available?

Reviewer #1: Yes

Reviewer #2: Yes

5. Is the manuscript presented in an intelligible fashion and written in standard English?

Reviewer #1: Yes

Reviewer #2: Yes

6. Review Comments to the Author

Reviewer #1: I see that the authors have implemented most of my recommendations. Overall, the manuscript now looks much completer and more understandable . I have only three comments that should be considered

1) This is just a small comment and is more about retrieving the authors changes. The lines the authors specified do not match either of the two uploaded versions of your manuscript. This makes retrieval very difficult.

2) The authors wrote that they have more information about Geweke a nd Gelman Rubin diagnostics in table S6. But this information is very incomplete. It is not enough to just write whether their model is convergent or not. It would be better to give the specifications. How many variables have been tested? How many of them w ere in the Gelman Rubin diagnostic above 1.05 (at best none). How many variables were unequal in the Geweke disgnostic and in which chains (at best less than 5%)? This information is important to measure the quality of Bayesian models using MCMC and must b e published.

3) My final point is again about missing resources. The authors wrote that they added a few sentences about this in line 641. This brings us back to the problem in 1): There is the b ibliography in this line in both manuscripts. Unfortunately, I could no t find this place at all. The easiest way would be to make a small chapter on this topic in the SI, because this is an important

point to classify the food reconstructions made.

Reviewer #2: Thank you for revising you manuscript. You have satisfactorily addressed my recommendations. I have now recommended the paper be accepted for publication.

7. PLOS authors have the option to publish the peer review history of their article (what does this mean?). If published, this will include your full peer review and any attached files.

Reviewer #1: No

Reviewer #2: **Yes: **Eric Guiry

---

## [Author Response · Author response to Decision Letter 1]

21 Aug 2023

We hopefully fulfilled all the requirements and provided all the information that was required from us by the editor and the reviewers. 

The reference list has also been reviewed.

---

## [Editor Report · Decision Letter 2]

4 Sep 2023

Long-term responses of Icelandic Arctic foxes to changes in marine and terrestrial ecosystems

PONE-D-23-03420R2

Dear Dr. Berthelot,

We’re pleased to inform you that your manuscript has been judged scientifically suitable for publication and will be formally accepted for publication once it meets all outstanding technical requirements.

Kind regards,

Dorothée Drucker

Academic Editor

PLOS ONE
---

## [Editor Report · Acceptance letter]

11 Sep 2023

PONE-D-23-03420R2 

Long-term responses of Icelandic Arctic foxes to changes in marine and terrestrial ecosystems 

Dear Dr. Berthelot:

I'm pleased to inform you that your manuscript has been deemed suitable for publication in PLOS ONE. Congratulations! Your manuscript is now with our production department. 

Kind regards, 

on behalf of

Dr. Dorothée Drucker 

Academic Editor

PLOS ONE